# Closing the Loop: Exploring Food Waste Management in the Near East and North Africa (NENA) Region during the COVID-19 Pandemic

Chedli Baya Chatti [1],*, Tarek Ben Hassen [2] and Hamid El Bilali [3]

1 Department of Social Sciences, College of Arts and Sciences, Qatar University, Doha 2713, Qatar
2 Program of Policy, Planning, and Development, Department of International Affairs, College of Arts and Sciences, Qatar University, Doha 2713, Qatar; thassen@qu.edu.qa
3 International Centre for Advanced Mediterranean Agronomic Studies (CIHEAM-Bari), Valenzano, 70010 Bari, Italy; elbilali@iamb.it
* Correspondence: cchatti@qu.edu.qa

**Abstract:** The COVID-19 pandemic disrupted global food waste patterns through unanticipated shifts in composition and quantities. This review explores the impacts of COVID-19 on food waste generation and management approaches in the Near East and North Africa (NENA) region during the recovery phase. This paper comprehensively explores food loss and waste in the NENA region. It presents a detailed analysis of pandemic-induced changes in household food waste behaviors, analyses the integration of circular economy principles in recovery strategies and policy implications, and outlines potential avenues for future research in this critical area. The key findings are threefold: First, this study reaffirms that food waste is a critical challenge in NENA, contributing to food insecurity, water scarcity, and environmental issues. Second, the pandemic catalyzed a dichotomy in consumer behaviors—panic buying initially increased waste, while hardship measures later encouraged sustainable waste reduction practices like meal planning and leftover use. Third, adopting a circular economy approach holds potential, yet its implementation remains limited in terms of curbing food waste and promoting sustainability in NENA. Overall, while the pandemic accentuated the urgency of tackling food waste, it also stimulated innovative policy thinking and strategic planning for building more resilient food systems. This paper concludes that leveraging pandemic-driven sustainability mindsets while addressing systemic drivers of waste will be key to mitigating food waste and its impacts moving forward. This paper offers timely insights into the evolving food waste management landscape in NENA, underscoring the need for integrated policies to navigate post-pandemic recovery effectively.

**Keywords:** food waste; COVID-19; circular economy; circularity; sustainability; waste recycling; waste reuse; Near East and North Africa (NENA)

## 1. Introduction

Food loss and waste (FLW) is defined as "a decrease, at all stages of the food chain, from harvest to consumption in mass, of food that was originally intended for human consumption, regardless of the cause" [1,2]. Food waste occurs at the downstream end of the supply chain, such as at the retail, food service, and household consumption stages. It refers to discarded edible food, regardless of whether it is still within the expiry date or has been left to rot [2–5]. Household food waste behavior may be influenced by several activities, including planning, grocery shopping, storage, cooking, and consuming [6]. As a result, a wide range of variables influences food waste, including behavioral (e.g., shopping and storage attitudes, insufficient knowledge), personal (e.g., level of education, experience, etc.), and product (e.g., packaging sizes, sale promotions, and discounts), as well as socio-economic characteristics (e.g., wages, food subsidies, etc.) [7].

The COVID-19 pandemic prompted a devastating international economic and financial downturn, negatively affecting socio-economic development and people's lives [8]. It tremendously influenced people's daily lives, including various substantial effects on diet and food-related practices, such as FLW [9]. The lack of proper storage facilities, transit disruptions, and labor shortages at the production level resulted in the loss of perishable products and excessive food waste in many countries [10]. At the consumer level, on the one hand, the pandemic significantly impacted household food shopping and planning, resulting in increased panic buying and food waste. On the other hand, some consumers began to adopt more sustainable behaviors, such as better food shopping planning and reusing leftovers to reduce food waste [11–13].

Due to several factors, including socio-economic development level and the efficiency of national healthcare systems, these habits are context-specific rather than global [14]. In the Near East and North Africa (NENA) region, which includes 20 countries spanning across North Africa and the Near East, the COVID-19 pandemic-related measures, as seen globally, led to many lifestyle changes and a shift in food habits. Even before the pandemic, FLW in the NENA region was a significant issue, leading to decreased food availability, exacerbated water crises and environmental challenges, and amplified food imports in an already heavily import-dependent region [15]. However, COVID-19's long-term economic recovery gives the NENA region a chance for sustainable growth by applying the circular economy approach. Certainly, the current socio-economic emergency triggered by the pandemic in the NENA region has generated unique opportunities for promoting a circular economy and circular resource management, which might stimulate economic recovery [16].

Moreover, the COVID-19 pandemic has highlighted the vulnerabilities of food systems in the NENA region and the importance of ensuring food security and sustainability. Understanding the impact of the pandemic on food waste and circular economy practices in the region is critical to developing effective policies and strategies to promote sustainable food systems and reduce food waste. However, there is a lack of research on the specific challenges and opportunities related to circular economy practices in the agrifood industry in the NENA region. The existing literature mostly focuses on developed countries, and there is a need for more research tailored to the unique socio-economic, cultural, and environmental contexts of the NENA region.

Consequently, this paper makes several valuable contributions to the academic literature on food loss and waste (FLW). First, it provides crucial new insights into the complex dynamics of FLW in the Near East and North Africa (NENA) region, specifically focusing on how the COVID-19 pandemic impacted food waste attitudes, behaviors, and quantities at the household level. Examining these pandemic effects within the unique socio-economic and environmental context of NENA represents a significant advancement, as this critical region has been understudied in existing FLW research. Second, this paper offers an original perspective on the relationship between global crises and localized FLW patterns, elucidating how disruptions caused by the COVID-19 pandemic influenced food waste in NENA households and food systems. This advances our theoretical understanding of the linkages between global events and regional FLW trends.

Third, this paper conducts an innovative assessment of how circular economy principles are integrated into post-pandemic recovery strategies and policies for curbing food waste in the NENA region. This analysis of policy and government responses provides practical guidance to policymakers on more sustainable solutions for managing FLW. Finally, the paper goes beyond analysis to propose tangible recommendations tailored to improving FLW reduction and management, specifically in the NENA context. This applied focus on actionable solutions contributes meaningfully to developing more resilient and sustainable food systems in the region.

In summary, by offering the first in-depth examination of pandemic-catalyzed changes in food waste dynamics and connections to policy in NENA, this paper significantly advances academic and practical understanding of FLW challenges and opportunities. The

insights from this timely research provide invaluable knowledge to guide policymakers, stakeholders, and researchers in building more sustainable food waste strategies as part of a green recovery in the NENA region and beyond. This paper makes pivotal contributions by elucidating global food waste transformations and corresponding policy responses to inform sustainable FLW management.

This paper is structured into seven main sections. Section 1 reviews the literature on the scale and characteristics of food loss and waste (FLW) in the Near East and North Africa (NENA) region, summarizing current knowledge on the magnitude of the issue. Section 2 outlines the study methodology, describing the data collection and analysis approach. Section 3 presents findings on attitudes, awareness, and self-reported behaviors related to household food waste during the COVID-19 pandemic. Quantitative and qualitative results provide insights into the pandemic's impacts on food waste awareness and actions. Section 4 uses empirical data to examine how the pandemic influenced food waste generation, composition, and quantities. Trends in household food waste levels and types are analyzed. Section 5 discusses integrating circular economy principles and food waste management strategies into post-pandemic recovery policies and initiatives in NENA countries. Policy documents are scrutinized to assess sustainability considerations. Section 6 summarizes this study's implications through a discussion and conclusion. Finally, Section 7 presents this paper's limitations and suggests promising directions for future research on this pressing topic.

## 2. Materials and Methods

The methodology of this study is based on two steps. The first step is a systematic review based on a search conducted on Web of Science (WoS) on 17 March 2023. The search utilized a specific query string: *("COVID-19" OR COVID19 OR Coronavirus OR "SARS-CoV-2") AND (food) AND (waste) AND ("Near East" OR "Middle East" OR "West* Asia" OR "North* Africa" OR Maghreb OR "East* Mediterranean" OR "South* Mediterranean" OR Arab OR Gulf OR Algeria OR Bahrain OR Egypt OR Iraq OR Jordan OR Kuwait OR Lebanon OR Libya OR Mauritania OR Morocco OR Oman OR Qatar OR Saudi OR Sudan OR Syria OR Tunisia OR "United Arab Emirates" OR UAE OR Yemen).* From this search, 14 documents were retrieved [12,13,17–28] and subsequently evaluated for their suitability based on two key criteria: geographical relevance (i.e., the document pertains to at least one country within the NENA region) and thematic pertinence (i.e., the document encompasses aspects of both COVID-19 and food waste issues). Consequently, only 10 documents [12,13,17–21,24–26] were eligible and included in this study.

Given the limited number of suitable scholarly documents obtained from Web of Science (WoS), the methodology was expanded to include a broader spectrum of the literature. Indeed, a further search was undertaken in other databases, including Google Scholar, which is renowned for its vast collection of non-indexed scholarly literature as well as grey literature. This supplementary search sought to include a broader range of information, such as reports, working papers, and policy briefs that are not commonly found in traditional academic databases but are important for understanding the full scope of COVID-19's impact on food waste issues in the NENA region. This inclusive method ensured that the research included a broader range of perspectives and facts, resulting in a more comprehensive understanding of the subject matter.

This extensive review process included a thorough examination of publications from international organizations, such as the United Nations, the Food and Agriculture Organization of the United Nations (FAO), the High-Level Panel of Experts on Food Security and Nutrition (HLPE), the World Bank, the World Food Program (WFP), the International Monetary Fund (IMF), and the United Nations Development Program (UNDP). Moreover, the research embraced contributions from regional bodies, notably the United Nations Economic and Social Commission for West Asia (ESCWA), to ensure a comprehensive understanding of the NENA region's unique context. This review was further enriched by insights from other research and policy institutions, such as the Economist Intelligence

Unit, the Centre for Strategic and International Studies (CSIS), and Wageningen University and Research.

In addition to scholarly and organizational sources, our methodology also incorporated valuable insights from leading regional newspapers and specialized news outlets, such as Arab News, Khaleej Times, The Peninsula Qatar, and Egypt Today, which were instrumental in providing real-time data, expert opinions, and ground-level perspectives on food waste challenges and initiatives within the region during the pandemic. These multiple sources gave an extensive viewpoint, which was critical for conducting a comprehensive analysis of food waste management dynamics in the NENA area during the COVID-19 pandemic.

## 3. Magnitude and Nature of Food Losses and Waste in the NENA Region

The NENA region is politically heterogeneous, with nations at various levels of economic development and unequal natural resources [29]. Food systems in the region are also marked by significant inequities, which may influence consumer choices and decisions and, consequently, food waste, in diverse ways. There are considerable income disparities between those in natural resource-rich countries (e.g., the Gulf Cooperation Council—GCC) and those in middle-income ones [29]. For instance, since GCC nations are capital-rich, food imports are exempt from financial restrictions. As a result, these countries have been less vulnerable to food price volatility than other food importers, and they have overcome local production shortfalls due to their solid fiscal position [11,30]. Consequently, in 2022, the Global Food Security Index ranked GCC countries as having the highest levels of food security globally and among Arab countries [31]. Moreover, food security vulnerabilities are more significant in conflict-affected countries such as Syria and Yemen [32,33].

Food waste is a major issue in the NENA region [15,34–40]. It leads to decreased food availability, exacerbated water crises, and several environmental consequences [37]. Food waste is chronically high in the NENA region, pointing to the inefficiencies, unsustainability, and inequality that characterize most agrifood systems in the region [41]. High food waste is not only uneconomical but also affects food security by increasing food imports in an already heavily import-dependent area due to the scarcity of natural resources [42]. According to the FAO [3], the share of lost or wasted food is about 34% across the NENA region. Several FAO reports [37,43] revealed that fruits and vegetables are the most wasted foods in the NENA region (in some cases, close to 50% of production), followed by fish and seafood (28%) and roots and tubers (26%). For example, research conducted in 2015 on post-harvest food losses in Morocco indicates that loss rates were much higher for fruits than soft wheat (Table 1).

**Table 1.** Estimation of food loss in wheat and fruit value chains in Morocco using survey and sampling methods, 2015 (percentage of physical losses).

| Product | Harvest | Storage | Transport | Wholesale | Retail | Processing |
|---|---|---|---|---|---|---|
| Soft Wheat | 10 | 14 | 1 | 10 | | 2 |
| Apple | 10 | 19 | 2 | 14 | 9 | --- |
| Citrus | 5 | 1–2 | 2 | 1–2 | --- | --- |
| Fig (fresh) | --- | --- | --- | 5 | | --- |
| Fig (dried) | --- | 2–5 | --- | Minimal | 2–5 | 5–10 |
| Prickly pear | 16 | --- | --- | --- | --- | --- |
| Dates | 14 | 19 | 2 | Minimal | --- | --- |

Source: FAO [44].

Nevertheless, apart from some rough estimates, there is a critical lack of accurate data on the causes and magnitude of FLW in the NENA region [37]. The FAO [3] estimates that per capita FLW is more than 200 kg/year in North Africa and West and Central Asia (cf. NENA region). To address the critical need for a more granular understanding of the extent and nuances of household food waste across the NENA region, the following table (Table 2)

provides a comparative analysis based on the numbers provided by the United Nations Environment Programme (UNEP) in the Food Waste Index Report of 2021 [5]. The table presents estimated household food waste per capita (kg/year) and total volume (tons/year) for nineteen nations in the area. It is worth noting that many countries in the region have very low or low confidence in these statistics, highlighting data collecting issues and the urgent need for more precise and thorough research. The outlier in the area is Saudi Arabia, the only country with a high confidence estimate, indicating a more robust data system than its counterparts in the region. This stark difference in data confidence levels across the region highlights not only disparities in data availability and reliability but also the need for more robust data collection and research to accurately assess the impact of household food waste and effectively tailor interventions.

**Table 2.** Comparative analysis of household food waste: a country-by-country overview of the NENA Region.

| Country | Household Food Waste Estimate (kg/Capita/Year) | Household Food Waste Estimate (Tons/Year) | Confidence in Estimate |
|---|---|---|---|
| **Algeria** | 91 | 3,918,529 | Very low confidence |
| **Bahrain** | 132 | 216,161 | Medium confidence |
| **Egypt** | 91 | 9,136,941 | Very low confidence |
| **Iraq** | 120 | 4,734,434 | Medium confidence |
| **Jordan** | 93 | 939,897 | Low confidence |
| **Kuwait** | 95 | 397,727 | Low confidence |
| **Lebanon** | 105 | 717,491 | Medium confidence |
| **Libya** | 76 | 513,146 | Very low confidence |
| **Mauritania** | 100 | 450,720 | Low confidence |
| **Morocco** | 91 | 3,319,524 | Very low confidence |
| **Oman** | 95 | 470,322 | Low confidence |
| **Palestine** | 101 | 501,602 | Low confidence |
| **Qatar** | 95 | 267,739 | Low confidence |
| **Saudi Arabia** | 105 | 3,594,080 | High confidence |
| **Sudan** | 97 | 4,162,396 | Very low confidence |
| **Syria** | 104 | 1,771,842 | Low confidence |
| **Tunisia** | 91 | 1,064,407 | Very low confidence |
| **United Arab Emirates** | 95 | 923,675 | Low confidence |
| **Yemen** | 104 | 3,026,946 | Very low confidence |

Source: UNEP [5].

According to more recent data [37], 32% of food waste in the NENA region happens at the consumer level and is concentrated in urban areas. In comparison, up to 68% occurs in the early phases of the food supply chain (production, transportation, manufacturing, and distribution/retail). The FAO [45] indicated that the drivers of FLW differ within the NENA area and consist of insufficient and poor infrastructure (e.g., cold chain, marketplaces) and inadequate regulatory and legislative frameworks. Baig et al. [46] highlighted that "the factors responsible for food waste include lack of awareness; insufficient and inappropriate shopping planning. Food waste in restaurants, celebrations, social events, and occasions is enormous. Waste is common in festivals and special events where the customs is to provide more food than required" (p. 1743). Therefore, according to Baig et al. [39], the most significant causes of food waste in the region are culture, food pricing, policy and industry issues, and awareness. Indeed, enormous food waste occurs on religious holidays, particularly in the fasting month of Ramadan, as well as on social occasions such as weddings and family reunions [37]. Food waste is a complex problem in the region, where consumers' and retailers' attitudes toward food waste are partly shaped by the region's unique culture, traditions, and history [42]. This assumption is corroborated by the results of surveys on household food wastage in different NENA countries such as Tunisia [47], Egypt [48], Algeria [49], and Morocco [50].

Moreover, residents in the NENA region, especially in GCC countries, take food for granted. This is due to the highly subsidized nature of food items and groceries, especially essential items such as bread, sugar, oil, and other staples. These subsidized items are widely available to all residents, contributing to a perception that food is abundant and easily accessible [51]. In the NENA region, governments have traditionally relied on subsidies to lessen the cost of food, primarily to protect the poor and share wealth [52]. In many countries in the region, governments subsidize bread and make it cheap to ensure societal access to basic foodstuffs. For instance, the Egyptian government spends over USD 3 billion annually on wheat imports, predominantly to support the country's long-running bread subsidy program. This program, known as the Tamween ration card system, provides subsidized bread to 73% of Egyptian households [53]. As a result, bread is both a critical source of nutrition and the most wasted food item. At the same time, recycling rates are low. In many parts of the region, only the informal sector carries out waste collection and recycling [54].

However, while the prevalence of subsidized food items in the NENA region might engender a perception of abundance and potentially contribute to increased food waste, it is imperative to consider the broader, more intricate socio-economic and cultural factors that play significant roles. As a matter of fact, the relationship between food abundance, prices, and waste is complex and not solely determined by subsidy schemes. For instance, subsidizing bread is often cited as contributing to high bread waste levels in the NENA region. However, it is critical to note that bread is the most consumed food and one of the most wasted foods in the world, even in many countries without such subsidy schemes, suggesting that other factors are also at play. Due to its brief shelf life and excess production, approximately 10% (900,000 tons) of bread is discarded throughout the supply chain, from production to consumer use [55]. For example, in the United Kingdom, 32% of household bread is wasted even without subsidies [56]. Similarly, over 50% of bread waste in Sweden happens at bakeries and retailers without subsidies [57]. Additionally, according to Jung et al. [58], wasted bread accounts for 13% of total food waste generated in Finland, 22% in the Netherlands, 23% in New Zealand, 27% in Norway, 7.9% in Portugal, 2.2% in South Korea, and between 12% and 17% in Sweden. The high levels of waste in these unsubsidized countries demonstrate that factors beyond subsidies, like cultural attitudes and retail practices, drive bread waste worldwide.

## 4. Food Loss and Waste during the COVID-19 Pandemic in the NENA Region

The COVID-19 pandemic affected global waste generation dynamics with unforeseen composition and quantity shifts, especially in food waste [59]. However, since it was caused by a human virus with no direct impact on agriculture production, the COVID-19 pandemic is not considered a direct food waste driver [60]. Consequently, FLW during the COVID-19 pandemic resulted from demand, supply, and logistics disruptions in the agrifood supply chain rather than direct food loss, as opposed to other epidemics (such as bird or swine flu), which caused direct food loss. Indeed, the pandemic influences food waste indirectly by causing changes in key food waste drivers [61].

At the production level, the lack of adequate storage facilities, transport interruptions, and a drop in demand in several countries led to losses of fresh items such as milk and vegetables. Labor shortages also affected crop harvesting [62]. Subsequently, social distancing measures and health checks at borders caused container delays, thus leading to the loss of perishable products and high food wastage [63]. For instance, in Iraq, vegetable farmers could not sell their products in local markets because of the curfew, resulting in spoiled produce and lost income [64]. Furthermore, limitations on mobility prohibited farmers from reaching markets, obtaining inputs, and vending their products, resulting in losses at the production level [63]. For instance, Jordan's severe local emergency plan prohibited farmers from accessing their fields, delaying daily operations and disrupting the harvest season. Likewise, in Tunisia, due to movement limitations, agricultural workers

had difficulty reaching their fields, and local markets faced a shortage of locally cultivated fruits [65].

At the consumer level, the pandemic considerably impacted household food buying and planning, including features identified as the leading causes of food waste in households [66]. However, the COVID-19 pandemic had paradoxical repercussions. On the one hand, panic buying was common in the early stages of the pandemic because of a lack of information about the virus and its severity. Stockpiling of non-perishable foods (such as flour, pasta, canned food, rice, etc.) increased in several countries in the region due to anticipated shortages in the near future [67,68]. In fact, panic buying and hoarding are prompted not by supply deficiencies but by consumers' concerns and anxiety over a prospective shortage [69–71]. Panic buying and stockpiling of food items was also triggered by fear and anxiety surrounding potential exposure to the COVID-19 virus. For example, a study conducted in Morocco discovered that most consumers reduced the number of food shopping visits they made and shopped less often than usual. To minimize the need for store visits and limit perceived risks of exposure to COVID-19 while shopping, consumers compensated by purchasing larger quantities and stocking up on more items during each trip (Table 3).

**Table 3.** Shopping behavior and purchasing changes during the COVID-19 pandemic in Morocco (*n* = 340).

| Variable | Statement | Frequency | Percentage |
|---|---|---|---|
| Shopping behavior change | I go shopping less than usual | 186 | 54.71 |
| | I go shopping like I used to | 102 | 30 |
| | I go shopping more than usual | 52 | 15.29 |
| Change in food purchase | I buy more than usual | 120 | 35.29 |
| | I buy the same as usual | 165 | 48.53 |
| | I buy less than usual | 55 | 16.17 |

Source: El Bilali et al. [72].

This obsessive purchasing behavior may have raised food prices, aggravated overconsumption (cf. obesity), and caused inadequate food availability [73]. Several countries, for example, experienced temporary shortages and price increases. These did not persist long since certain governments soon intervened to stabilize prices via numerous strategies [67]. Moreover, those who have the financial wherewithal to purchase more food may end up hoarding it, causing havoc among underprivileged residents [74] and depriving some vulnerable groups (e.g., the elderly or the poor) of accessing particular food products [75]. Due to storage constraints, inappropriate cooking techniques, or the overpreparation of meals, panic buying may result in increased household food waste, particularly for fresh items. Excessive purchases caused by panic buying imply that more food will spoil before it can be eaten [9,76].

On the other hand, the COVID-19 pandemic opened up hitherto unforeseen opportunities for a more conscientious approach to consumption [77]. During the pandemic, several consumers in the region adopted a thriftier attitude and reduced food waste. Indeed, several studies conducted in the region revealed that household food waste decreased. Additionally, more and more people were cooking and baking at home due to the closing of the HORECA channel (hotels, restaurants, and caterers). For instance, a survey of 284 people in Tunisia revealed that about 89% of the participants indicated being aware of food waste. It was shown that COVID-19 measures led to better food shopping habits and an overall reduction in food waste. Also, 85% of those polled said they would not waste any food they purchased, and the majority outlined that they had a system for storing and consuming any leftovers [9]. Despite widespread panic shopping, there was no rise in food waste in Lebanon [78]. Since COVID-19 became a significant problem in the country, 73% of respondents said they had stockpiled food. However, 86% said they were not wasting more food, and 80% said they were more conscious of how much they were wasting. Likewise,

in Qatar and Oman, food waste dropped due to the absence of panic buying and food stockpiling [13,24] (Figure 1).

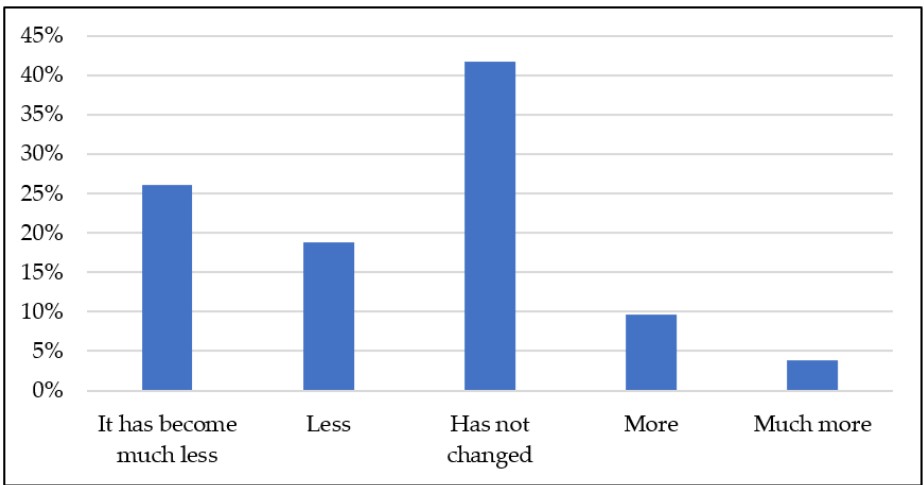

**Figure 1.** Food waste change during the COVID-19 pandemic in Qatar. Source: Ben Hassen, et al. [13].

Food waste also decreased in Qatar during the Omicron variant wave, with 76.17% of survey respondents reporting that they had reduced their food waste since the start of the wave and 68.82% reporting that they were more aware of the amount of food waste [79].

These results may be due to several factors. Firstly, the hospitality and tourism sectors are the two most prominent causes of food waste in the region [80]. COVID-19 had a substantial negative impact on the hospitality industry [81]. Lockdowns and social distancing regulations imposed the closure of the HORECA channel. Also, the closing of international borders halted incoming tourism. Consequently, many countries in the region, especially the GCC, saw decreased food waste in 2020 [82]. Secondly, family reunions became less frequent due to social distancing restrictions, reducing the waste caused by laying out massive spreads during holidays and social events such as Ramadan or weddings [82]. For instance, usually, during Ramadan, 30–50% of the food produced in Saudi Arabia is thrown away; this figure stands at 25% in Qatar and 40% in the UAE [83].

Thirdly, across the region, numerous good food management practices were adopted by households throughout the pandemic, including increased pre-shop planning, improved at-home food storage, and innovative ways of cooking/preparation (e.g., batch cooking and using up leftovers) [9]. Many households began purchasing more selectively and conserving what they could not finish for later [82]. Adopting these practices is driven by various motives, including a desire to save time and money, a desire to prevent running out of food, and a deep suspicion of grocery stores. Because individuals spent more time in their homes, they could spend more time in the kitchen without feeling pressured [84].

Furthermore, according to a poll of 200 participants from 10 West Asian nations conducted between July and November 2020, the use of leftovers changed following the pandemic. Undoubtedly, when questioned about the use of leftovers in their households, the proportion of respondents who stated "regularly/frequently" rose between the pre-COVID (42%) and post-COVID (50%) periods (Figure 2). This might have been due to an increased focus on healthy eating to promote immunity, since home-cooked meals are often considered more beneficial. This would encourage individuals to save leftovers for later instead of throwing them away [83].

However, according to Jribi et al. [9], the socio-economic circumstances of the pandemic (i.e., food supply, restricted mobility, and loss of income) are more likely to have influenced customer behavior towards food waste than a pro-sustainability attitude. Aside from being a public health concern, COVID-19 was a far more severe issue. The pandemic also triggered a global financial and economic crisis, leading to rising unemployment and widespread poverty [85]. Due to necessity, consumers reduced their food waste during the

crisis. When faced with hardships, people tend to save rather than discard food, resulting in a noteworthy decrease in waste generation. These difficult circumstances compelled consumers to waste less food out of need [86], as observed during previous recessions in Greece and Italy [87,88] and during the ongoing economic crises in Lebanon [89] and Iran [17]. Nevertheless, the factors that aided in minimizing food waste during the COVID-19 pandemic were temporary and may not endure now that these measures have been abandoned and consumers have returned to their former habits and behaviors.

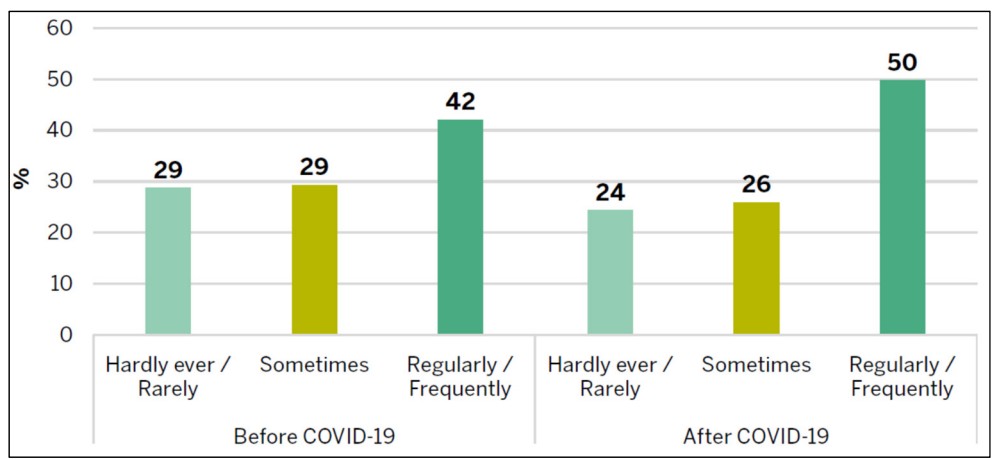

**Figure 2.** Recycling of food leftovers in West Asia pre and post COVID-19. Source: UNEP [83].

## 5. The Circular Economy and Food Waste in the NENA Region: Potential and Challenges

Long-term economic recovery from COVID-19 provides the NENA region with an exceptional chance for sustainable development [16]. While environmental preservation is often deprioritized, the present socio-economic crisis triggered by the pandemic in the NENA region has created a unique potential for fostering a circular economy and circular resource management, which might catalyze economic recovery. Circular business models use local resources to minimize import reliance and diversify suppliers for increased resilience [90]. Further, putting circularity at the center of natural resource plans will increase resource availability and value extraction. According to Al-Saidi et al. [91], it is possible to solve environmental challenges while simultaneously ensuring long-term supply security in the water and food sectors by promoting consumer awareness and resource circularity. The main goal is to "close the loops" by promoting reuse and recycling via economic structures. Circularity has the potential to be highly useful for arid regions, and it is typically hailed as a means to preserve resources while still creating prosperity [92,93].

Aside from immediate economic gains, circularity provides additional concrete and significant advantages that are incredibly relevant to the current NENA context, such as food waste. Indeed, addressing food waste with a circular mindset helps to reduce or reuse the volume of surplus food generated [94]. Indeed, on the one hand, circular economy approaches would reduce pollution and waste in the region, improving health and environmental circumstances. On the other hand, they would bring new economic possibilities to local populations, contributing to the region's political stability and economic diversification [95].

More recently, in the NENA region, we noticed a growing awareness of the importance of addressing the issue of FLW at the level of public institutions and policy, civil society and NGOs, and the private sector. Firstly, policymakers in the region have started to recognize the importance of the circular economy and reducing food waste. For instance, at the policy level, in January 2021, the Cabinet of the United Arab Emirates (UAE) Government approved the Circular Economy Policy (2021–2031), a comprehensive plan identifying the country's framework for sustainable governance and better usage of natural resources, by implementing consumption and production processes that assure the quality

of life for present and future generations. The strategy has many important goals, including improving environmental health, assisting the private sector in implementing clean manufacturing techniques, and lowering natural environmental stress to accomplish the country's ambition of becoming a worldwide leader in green development [96]. Reducing food waste is considered a priority in the strategy. Indeed, the strategy states, "By adopting circular economy strategies in the food sector, the UAE expects that its ecosystems will be healthier, its food healthier and more nutritious, its food wasted reduced and its organic wastes more productively used." (p. 15). The strategy also aims to raise public awareness and implement educational campaigns focused on nutritional guidelines and food waste reduction. These efforts would inform consumers about correctly interpreting best-before and use-by labels to decrease unnecessary waste. Outreach related to social events like weddings and religious celebrations would further promote less wasteful food habits. The strategy seeks to spread knowledge on mitigating food waste [97].

Likewise, in Qatar, in August 2021, the Ministry of Municipality and Environment announced a resolution on food waste. All outlets must now sort their garbage into solid and organic categories. The Ministry also established a strategic plan to eliminate agricultural overproduction and food waste. Due to this endeavor, organic waste will also be recycled into fertilizers and fodders. According to the plan, the school curriculum will include lessons on food waste reduction and public awareness campaigns will be created to promote a food-saving mindset. These programs will also be conducted in conjunction with restaurants to teach them how to prepare and serve meals that meet the needs of their consumers [98].

Moreover, there was a discussion of new legislation in the Egyptian House of Representatives in May 2022 to regulate food waste and encourage its redistribution, recycling, and donation. Food service providers such as restaurants and grocery stores face various fines under the new rule if they do not donate food that is fit for human consumption. The policy encourages and incentivizes recycling and a more reasonable approach to food consumption to prevent wasting surplus food people cannot consume [99].

In addition, food waste might be reduced with the help of startups and digital technologies [90]. Recently, many startups have been created in the NENA region to address the issue of food waste. For instance, in 2021, a food waste and climate change app, EroeGo, was created in the United Arab Emirates. EroeGo is an online grocery marketplace that allows consumers to purchase fresh products that are about to expire at reduced prices [100]. The startup also aims to change the notion of "unwanted" food in the region by providing a transparent platform that offers its consumers basic nutritional information, an efficient purchasing cycle for fresh items, and fair remuneration for its delivery drivers. Likewise, the startup Foodeals, created in Morocco in 2020, intends to become the greatest anti-food waste movement in the Middle East and Africa. Users may geolocate themselves and discover nearby companies providing specials or unsold items of the day using its application, inspired by the circular economy. Thus, the customer saves money, and the shopkeeper makes up for their loss of income while increasing their exposure. Currently functioning in Fez (central Morocco), Foodeals plans to expand to the Kingdom's other main cities shortly, with a dozen pilot firms already in place. As part of its business-to-business (B2B) strategy, the startup aims to connect significant supermarkets and agribusinesses with NGOs with daily food requirements [101]. In the United Arab Emirates, Winnow, a British startup, makes kitchens smarter to aid food service and hospitality businesses. Artificial intelligence tools enabled by Winnow are used by some of Dubai's most prestigious hotels to identify the kind of food items that are thrown out, as well as the amount of food that is thrown away. To reduce food wastage, commercial kitchens might use the information provided by the data to monitor their purchases and menus [102].

## 6. Discussion and Conclusions

Addressing FLW drivers throughout value chains gives us a chance to address some of the key issues within the NENA region's agrifood systems and contribute to goals such

as increasing income and employment, enhancing access to healthy food, and minimizing climate change impact, as well as achieving a better use of finite natural resources, notably arable land, and water [41]. With unexpected changes in content and amount, the COVID-19 pandemic significantly influenced worldwide FLW generation patterns, particularly food waste. This paper highlights that the current socio-economic crisis triggered by the pandemic in the NENA region has generated unique opportunities for creating a circular economy and circular resource management, which might stimulate economic recovery. Further, putting circularity at the center of natural resource management plans will increase resource availability and value extraction. Consequently, in the NENA region, we notice a growing awareness of the importance of addressing the issue of FLW at all levels (regional, national, and local) and from different stakeholders (from the public, civil society, and private sectors).

The previous section highlighted some promising developments related to circular economy and food waste reduction emerging in a few NENA countries. It is important to acknowledge that circular economy strategies in the region are still in their infancy, especially when it comes to concrete policies and national plans targeting food loss and waste. Consequently, few countries in the NENA region have designed and implemented strategies or policies promoting a circular economy. This study highlighted individual cases from the UAE, Qatar, and Egypt. However, there is no comprehensive regional circular economy policy or plan for reducing food waste. Specific themes, such as increasing home waste sorting, school education campaigns, and regulating food donations, have been explored on a national or municipal level rather than as part of a coordinated regional effort.

Based on a recent report by the Deutsche Gesellschaft für Internationale Zusammenarbeit (GIZ) (German Society for International Cooperation) [103] examining the condition of circular economy in seven countries within the region (Algeria, Egypt, Jordan, Lebanon, Morocco, Tunisia, and the Palestinian Territories), it is evident that the CE encounters several challenges in this area. Firstly, the interpretation of the CE in these countries primarily focuses on steps aimed at mitigating the effect of current activities rather than adapting to reduce the impact of new activities from the start. In reality, the MENA area seems deficient in material circularity-focused CE projects that aim to close the loop by concentrating on the entire life cycle of products and services. Second, although the governments of these countries have developed national waste management policies, their actual enforcement and execution are limited, reducing their overall effectiveness. In certain countries, such as Morocco, strong legal frameworks have been put in place, and specialized government organizations have been entrusted with aiding the transition to a circular economy.

On the other hand, countries such as Algeria, Lebanon, the West Bank, and Gaza have challenges in moving these efforts forward, principally due to political instability and shaky governance institutions. Finally, research on CE themes in the MENA area is limited and focused on specific industries for each nation, with Morocco and Tunisia being the most active. Most of the available literature comes from foreign organizations rather than local research institutions, and it lacks information on themes like circular design and innovation [103].

As a general observation, short-term post-pandemic recovery strategies in the region encouraged business as usual, and most countries did not aggressively pursue green, resilient, or merely longer-term recoveries in the immediate aftermath of the pandemic [16]. However, given the future issues the region will face, such as climate change and rising population, strengthening circularity becomes even more imperative. While the circular economy can potentially reduce food loss and waste in NENA, the actual implementation is still in its early stages. More studies are required to evaluate the policies and initiatives in all NENA countries comprehensively. Standardized objectives and measuring methodologies would help nations learn from one other's circular economy projects for reducing food waste. Regional collaboration might also aid in the rapid adoption of circular processes. As a result, policymakers must consider how best to encourage circular economy principles in the public and private sectors [95].

To solve these issues, addressing the distorting effects of food subsidies, the lack of community engagement, the inadequacy of current policies, and the inadequate channels for coordination between the many sectors engaged in reducing food waste is compulsory. Subsidies are financially unsustainable for governments, yet they are socio-economically necessary to maintain social peace [104]. Further, the impact of the present conflict in Ukraine on global food systems and supply chains (e.g., price increases) may provide a unique chance to increase awareness of the necessity of reducing food waste to support food security in the NENA region. The war emphasizes that the shift towards more sustainable and resilient food systems that guarantee food and nutrition security in the face of crises is timely, urgent, and highly needed. Given the interconnected nature of global agricultural markets, the conflict between Russia and Ukraine, two major players in the global food and fertilizer industries, has provoked extensive concerns regarding food security worldwide and sustaining global food supplies [105,106]. The conflict challenges many countries, particularly food import-reliant NENA countries [107,108].

Finally, reducing food waste and promoting the circular economy in the NENA region require collaborative efforts between different stakeholders, effective policies and regulations, innovation, and a more holistic and systemic approach that operationalizes connections between the circular economy and other alternative economic models. Moving from fragmented efforts to comprehensive food waste reduction roadmaps supported by solid monitoring and evaluation systems is required. A circular economy approach that prioritizes renewable energy, minimizes waste, and optimizes resource use can contribute to environmental sustainability. In contrast, a green and blue economy approach can promote sustainable economic growth and enhance social welfare. By adopting these strategies, the NENA region can achieve a sustainable food system that benefits both the environment and society [109].

First, collaborative efforts and networking between governments, the business sector, consumer groups, and non-governmental organizations are essential to meet the challenges the agrifood industry poses. This collaborative approach can promote the exchange of ideas and best practices, enabling the sharing of resources and building partnerships to address the issue of food waste. Policies and regulations play a significant role in driving measures to reduce FLW, but their effectiveness depends on faithful execution and mechanisms to verify compliance. The Food and Agriculture Organization (FAO) emphasizes the importance of implementing policies and regulations that encourage responsible production and consumption patterns to reduce food waste [37].

Second, innovation is crucial to reducing food waste and promoting circularity and circular economy in the food system in the NENA region. Technological innovations such as developing efficient packaging and storage systems can minimize food spoilage and waste. Social innovations such as education and awareness campaigns that encourage responsible consumption and discourage food waste can also reduce food waste. Likewise, efforts should be made to increase awareness among residents about the environmental, social, and economic consequences of food waste. Launching food waste awareness programs tailored to cultural contexts, focusing on families, and training food industry actors is required. Messaging should encourage better buying habits, storage methods, portion proportions, and waste separation, which can help change consumer behavior and promote responsible food consumption and waste reduction practices. Third, efforts should be made to encourage and control waste reduction in the retail sector via training, certification programs, and legal restrictions on discarding unsold edible food.

Fourthly, future policies and initiatives addressing food waste in the NENA area should expressly include and recognize the informal sector's contribution. The informal sector, primarily involved in garbage collection and recycling, is a largely neglected resource for achieving the most significant waste reduction and prevention goals. This sector's extensive knowledge of local waste management techniques, agility, and community involvement make it an ideal partner in building a more robust and efficient circular economy. To realize this potential, strategic actions should include extensive research to

document the sector's current practices and challenges, policy integration to formalize and regulate operations, capacity-building initiatives to improve worker skills, stakeholder collaboration for innovative waste management solutions, and the implementation of incentive mechanisms to encourage active participation in food waste reduction initiatives. Adopting these strategic activities will not only address the environmental and economic aspects of waste management, but will also promote social inclusion and equality. Moreover, the countries of the region should develop national food loss and waste reduction objectives that are connected with Sustainable Development Goals (SDGs) and implement food waste taxes to promote change.

Finally, given the lessons learned from the COVID-19 pandemic, it is critical to understand food waste management not as a separate issue but as an important component of more extensive public health emergency preparation and response. The pandemic has highlighted our food system's vulnerability to global health crises, emphasizing the need of robust and adaptable food waste management measures. The lessons learned during this era provide essential information for future situations, implying that food waste management should be included in public health emergency preparedness to maintain food security, sustainability, and community well-being. This study adds to our knowledge of the complex interaction between food waste management and crisis response mechanisms by putting our results in the context of public health catastrophes. It is believed that these findings will help policymakers and stakeholders create resilient systems capable of overcoming the multiple difficulties offered by future global crises.

## 7. Limitations and Future Directions

The main limitation of this paper is that it focuses specifically on the NENA region. As a result, our findings and recommendations may have limited generalizability to other areas or contexts. The specific socio-economic, cultural, and environmental characteristics of the NENA region may influence the applicability of the suggested strategies in different geographical areas. Future research on sustainable food waste management in the NENA region can focus on the following directions to further advance knowledge and address existing gaps. Firstly, conducting in-depth studies that delve into the specific socio-economic, cultural, and environmental contexts of individual countries or sub-regions within the NENA region would provide a more nuanced understanding of food waste management practices. Such research would help identify region-specific challenges, opportunities, and potential solutions tailored to local contexts.

Secondly, although the preliminary results indicated that COVID-19 hardship measures decreased household waste in the short term, longitudinal research is required to determine long-term attitudinal and behavioral changes. Monitoring whether enhanced planning and storage practices remain after the pandemic gives essential insights into social habit formation processes. Follow-up surveys may reveal if the increased waste awareness is a temporary crisis reaction or an incentive for a proper move to sustainability.

Thirdly, studies should be conducted to assess the efficacy of context-specific behavioral nudges and social marketing approaches in encouraging home waste reduction. Environmental triggers, customized messages, and social comparisons are all approaches that should be tested and refined depending on cultural variations. Similarly, analyzing educational efforts may help to optimize welfare and sustainability results.

Finally, more research is needed to assess the effectiveness and impacts of policy interventions, regulations, and governance frameworks to reduce food waste in the NENA region. Comparative studies analyzing the outcomes of different policy approaches can inform evidence-based decision-making and support the development of robust policies that encourage sustainable practices. Community-engaged research may also explore the feasibility and acceptability of suggested policy initiatives to increase their acceptance. It is critical to foster such research–policy cooperation to transform academic knowledge into significant societal shifts.

**Author Contributions:** Conceptualization, C.B.C. and T.B.H.; methodology, H.E.B.; validation, C.B.C., T.B.H. and H.E.B.; formal analysis, C.B.C., T.B.H. and H.E.B.; writing—original draft preparation, C.B.C.; writing—review and editing, C.B.C., T.B.H. and H.E.B.; supervision, C.B.C.; project administration, C.B.C.; funding acquisition, C.B.C. All authors have read and agreed to the published version of the manuscript.

**Funding:** Open Access funding provided by Qatar National Library.

**Institutional Review Board Statement:** Not applicable.

**Informed Consent Statement:** Not applicable.

**Data Availability Statement:** No new data were created or analyzed in this study. Data sharing is not applicable to this article.

**Conflicts of Interest:** The authors declare no conflicts of interest.

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
