# Peer review of "Closing the Loop: Exploring Food Waste Management in the Near East and North Africa (NENA) Region during the COVID-19 Pandemic"

_sustainability, doi:10.3390/su16093772_

Round 1
Reviewer 1 Report
Comments and Suggestions for Authors
This review paper's objective 1 is to present the magnitude and nature of food losses and waste in the NENA region. Then, objective 2 of the paper is to examine pandemic-induced changes to food waste attitudes, awareness, composition, and quantities. Lastly, in objective 3 the paper discusses how recovery policies and initiatives in NENA countries address the circular economy and food waste management. In brief, this paper aims to elucidate pandemic-catalyzed changes in NENA region food waste dynamics and connect these transformations to contemporary food waste management and recovery policy.
However, the review paper did not follow either a scope comprehensive review (not all NENA region countries' food waste was covered) or a meta-analysis systematic review to address objectives 1 & 2. I recommend the authors use a systematic review process for objective 1 and cover with further detail examples and case studies from all NENA region's countries. Meanwhile, use the scope review process to address objective 2 of the study.
I will provide a further detailed review if such major changes are addressed.
Comments on the Quality of English Language
If the major chnages are considered by the authors, I will provide comments/review of the English quality.
Reviewer 2 Report
Comments and Suggestions for Authors
Overall a well written paper, with good structure and approach.
There are some points of attention that address critical thinking and the way results are presented across the NENA region.
Considering the link between abundance - food prices - food waste; there is very little scientific evidence that these are causally related, albeit correlations can be found. E.g. the Tamween ration is presented to support bread being the most wasted food item. Bread is very typically one of the highest wasted products, also in countries without such a subsidezd ration scheme. The authors could substantiate this section. this critical note should be part of discussing potential differences between food waste drivers and how they relate with specific national or regional circumstances. Also, because there is very little scientific evidence that connects specific food products with 'unique' reasons of why it's wasted.
The authors touch upon informality / informal sector as an agent in the FLW reduction/prevention area, but this is not substantiated with further evidence. If there would be differences with e.g. EU context, the magnitude & 'structure' of the informal sector has a much larger role in the NENA region: this will also impact policy making & intervention development. The authors could include a comment on this in explaining the scoping of the research (questions/methodology) or use it in the disucsison section as well.
The authors dutifully note the FAO stats on FLW in the Region, stating there are inaccurcies, It does not become clear if they undertook a specific search into the 20 countries' FW reporting practice / availablity of data. in policy development, baseline information is very important and the availablity of reliable/high quality/granular data is paramount.
The authors could be more strict in scoping either food loss or food waste, or include both 'types'. they use the definition of food waste stated in the intro, but continue on including the full supply chain. this is inconsistent. In the same line of thought, table 1 header mentions losses only, and then includes wholale/retail segments as well (which should be omitted, or the header be changed into FLW). there is a LOT of discussion on defining FLW, I am not particular about which one is used, but the one presented should be consistenly followed throughout the study so the terminology is used correctly.
The evidence brought to the table on NENA food waste amounts would expect to deliver an overview table where all 20 countries are compared. But this is not included, instead, anecdotal examples are mentioned. Also, would West-Asia be in the NENA region? (just chekcing here).
The analysis of CE continues to be 'anecdotal', instead of scrutineously comparing the 20 countries, A characterisation or classificaiton would help to understand where differences sit, and how these could be explained. e.g. Do all 20 have a defined target / measurement approach / action plans developed etc. including an assessment of which topics of actions are selected (e.g. focus on sorting household waste, school programs, etc.).
Reviewer 3 Report
Comments and Suggestions for Authors
1. The Abstract should underline in short the main results. The structure of the paper is described, but it would be more relevant for the authors to end this part of the article with one or more observations derived from the research undertaken, which deserve to be emphasized.
2. The paper should be better organized. Although it is defined as a review paper, it looks like a newspaper article. The information is not well organized to highlight the achievement of the proposed objective.
3. The paper objective declared, still stated in the Abstract, is not clearly pursued, but is mentioned among other issues.
4. The article should be restructured in a scientific manner, with a clearer mention of the methodology used and the contribution of the work to the development of this area.
Comments on the Quality of English Language1. There are some minor spelling errors.
2. Please avoid repeating too frequently some expressions like ”Indeed”, or furthermore/further...
Reviewer 4 Report
Comments and Suggestions for Authors
Dear researchers,
Please pay more attention to the used methodology. The information presented in the research needs improvement.
Reviewer 5 Report
Comments and Suggestions for Authors
This paper elucidated pandemic catalyzed changes in NENA food waste dynamics and connect these transformations to contemporary food waste management and recovery policy. I think this paper is an interesting topic. But it needs a minor revision before accepted. The following changes are recommended to help improve things:
1. Abstract should introduce research findings rather than elaborate on the paper's structure.
2. The paper addresses the issue of food waste management during the COVID-19 pandemic. However, based on existing analyses, it appears that only food consumption is directly linked to the pandemic, while other analyses may be unrelated to COVID-19. In other words, once the pandemic concludes, what is the general significance of this study? It is suggested that the author approach the topic more broadly, analyzing food waste management issues from the perspective of public health emergencies. Using COVID-19 as an example may provide the research with more general applicability.
3. In addressing the knowledge gap in the introduction, the focus should be on academic contributions rather than the uniqueness of specific regions.
4. There may be an excess of website citations in the references, which might not be entirely appropriate.
Comments on the Quality of English LanguageThe language expression is generally smooth.
Round 2
Reviewer 3 Report
Comments and Suggestions for Authors
The article has clearly been improved with better structure and coherence.